# Preparation and Characterization of a Janus Membrane with an “Integrated” Structure and Adjustable Hydrophilic Layer Thickness

**DOI:** 10.3390/membranes13040415

**Published:** 2023-04-06

**Authors:** Ruixian Zhang, Chengyu Deng, Xueyi Hou, Tiantian Li, Yanyue Lu, Fu Liu

**Affiliations:** 1Guangxi Key Laboratory for Polysaccharide Materials and Modification, Guangxi Higher Education Institutes Key Laboratory for New Chemical and Biological Transformation Process Technology, School of Chemistry and Chemical Engineering, Guangxi Minzu University, Nanning 530006, China; 2Ningbo Institute of Materials Technology and Engineering, Chinese Academy of Sciences, Ningbo 315201, China; 3School of Materials Science and Engineering, Hebei University of Science and Technology, Shijiazhuang 050018, China

**Keywords:** polyvinylidene fluoride, hydrophilic modification, Janus membrane, oil-water emulsions, separation

## Abstract

Oil-water emulsions are types of wastewater that are difficult to treat. A polyvinylidene fluoride hydrophobic matrix membrane was modified using a hydrophilic polymer, poly(vinylpyrrolidone-vinyltriethoxysilane), to form a representative Janus membrane with asymmetric wettability. The performance parameters of the modified membrane, such as the morphological structure, the chemical composition, the wettability, the hydrophilic layer thickness, and the porosity, were characterized. The results showed that the hydrolysis, migration, and thermal crosslinking of the hydrophilic polymer in the hydrophobic matrix membrane contributed to an effective hydrophilic layer on the surface. Thus, a Janus membrane with unchanged membrane porosity, a hydrophilic layer with controllable thickness, and hydrophilic/hydrophobic layer “structural integration” was successfully prepared. The Janus membrane was used for the switchable separation of oil-water emulsions. The separation flux of the oil-in-water emulsions on the hydrophilic surface was 22.88 L·m^−2^·h^−1^ with a separation efficiency of up to 93.35%. The hydrophobic surface exhibited a separation flux of 17.45 L·m^−2^·h^−1^ with a separation efficiency of 91.47% for the water-in-oil emulsions. Compared to the lower flux and separation efficiency of purely hydrophobic and hydrophilic membranes, the Janus membrane exhibited better separation and purification effects for both oil-water emulsions.

## 1. Introduction

Oil wastewater, formed by oil entering the ecological water system through different channels, is a type of wide-range, harmful pollution that leads to catastrophic effects for natural ecosystems and may cause harm to human health [1,2,3,4]. Oily wastewater can generally be divided into two types based on the oil droplet size: hybrid and emulsified [5]. The latter causes the oil phase to be evenly and stably distributed in the aqueous phase, owing to a large amount of surfactant; thus, it requires a separation treatment that combines multiple methods [4,6]. Membrane-filtration technology [7,8,9] is considered a prominent candidate technology for the separation of stable and dispersive oil–water emulsions, owing to its high efficiency, low cost, easy extensibility, and other features. Hydrophobic [10,11] and hydrophilic membranes [12,13,14] can separate oil-water emulsions unidirectionally. However, Janus membranes are characteristic separation membranes with opposite or different surface-wetting properties on both sides of the membrane; thus, they can perform oil-in-water (O/W) or water-in-oil (W/O) emulsion switching separation [15,16,17,18]. Additionally, Janus membranes exhibit wettability specificity for liquid transport [19,20], membrane distillation [21,22], and fog collection [23].

Most existing research on Janus membranes involves the post-modification of porous matrix membranes to prepare hydrophilic–hydrophobic double-layer or multilayer composite membranes. Single-sided coating/spraying/deposition [24,25,26], surface grafting [27,28], hydrogel [29,30], and electrospinning [31,32,33] have been used to construct hydrophilic (hydrophobic) coatings on the surfaces of other hydrophobic (hydrophilic) substrate membranes. In a study by Yang et al. [26], a polydopamine/polyethyleneimine hydrophilic solution was deposited in multiple cycles on the inner lumen side of a hollow fiber that was originally hydrophobic to create a serviceable hydrophilic coating. However, the pores of the inner wall of the hollow fiber membrane were covered to a certain extent, and the results showed that the hydrophilic coating was gradually lost as the practical application time increased. A hydrophilic hydrogel coating was produced by spraying a chitosan-perfluorooctanoate suspension containing silica nanoparticles onto a hydrophobic polyvinylidene fluoride (PVDF) substrate [29]. However, the initial flux of the modified membrane was significantly lower than that of the original membrane because the hydrogel coating blocked the membrane pores. Similarly, Zhu et al. [33] constructed a superhydrophobic hybrid polymer layer and a superhydrophilic microsphere string-structured fiber layer on a PVDF nanofibrous membrane substrate using continuous electrospray technology. The results of the membrane application showed that the loading of too many nanoparticles blocks the gap between the membrane fibers, thereby decreasing the membrane flux. The above studies effectively constructed Janus membranes with asymmetric wettability; however, some unavoidable challenges, such as complicated steps, clogged membrane pores, and weak adhesion of the modified layers, are not conducive to the long-term application of Janus membranes.

This study makes a unique contribution to the literature by using the unidirectional migration and thermal-crosslinking polymerization of high-molecular hydrophilic polymers, namely poly(vinylpyrrolidone-vinyltriethoxysilane) (PVP-VTES), to complete the construction of an available hydrophilic layer inside the PVDF matrix membrane. The synergistic effectiveness of the low surface energy of the PVDF matrix and the micro/nanostructure caused by the peeling of the supporting body enabled the membrane bottom surface to maintain a favorable hydrophobicity. The advantages of this modification method are as follows. First, the final composite membrane with asymmetric wettability is a “structural integration” of the hydrophilic and hydrophobic layers; that is, the membrane exhibits no internal structure delamination. Second, the hydrophilic layer thickness of the composite membrane is adjustable. Most importantly, the blockage of the membrane pores and decreased porosity, which are challenges that other post-modification methods face, are avoided by this modification strategy. The Janus membrane was applied to switchable separation experiments on O/W and W/O emulsions, and a steady flux and improved separation effect were obtained.

## 2. Materials and Methods

### 2.1. Materials

PVDF (Kynar 761-A) was supplied by Arkema (Guangdong Agent, Guangdong, China). *N*-vinyl-2-pyrrolidone (99%), 2,2′-azobenzene (2-methylpropyonitrile) (98%), Congo red, Sudan III, brilliant blue, sodium dodecyl sulfate (SDS, ACS, ≥99.0%), and span80 were all purchased from Aladdin Reagent Co., Ltd., Shanghai, China. VTES and triethyl phosphate (TEP) were obtained from Sinopharm Chemical Reagents Co., Ltd., Beijing, China. Polyester non-woven fabric (PET, density = 0.8 g·cm^−3^) was obtained from Shanghai Jiujun New Material Technology Co., Ltd., Shanghai, China. Soybean oil was used as the oil drops in the oil-water (O/W, W/O) emulsion feed solution. Deionised water was used in this study, and all the reagents were used directly without further purification.

### 2.2. Method Used for Membrane Preparation

#### 2.2.1. Hydrophobic Membrane Preparation

PVDF with a mass fraction of 15% was weighed, added to the solvent (TEP), and stirred at a constant temperature (80 °C) until a transparent and colorless homogeneous casting solution was observed. The defoaming pouring solution was evenly scraped onto the PET support and then immediately transferred into a mixed coagulation bath (TEP: H2O = 1:1) for 30 s to complete the solvent–non-solvent phase transformation and take the shape of a membrane. The membrane was again shifted and immersed in deionized water for 12 h to completely remove the residual solvent. Finally, the membrane was removed and dried naturally to obtain a PVDF hydrophobic microporous membrane, denoted as M0.

#### 2.2.2. Preparation of Janus Membrane

The hydrophilic modifier was formed by mixing the pre-synthesised hydrophilic copolymers (PVP-VTES) [34,35] with deionized water in a 1:1 ratio. The chemical reaction representing the synthesis of PVP-VTES is shown in chemical reaction (1). The hydrophobic membrane (M0) was completely immersed in the modifier with its surface up and allowed to stand for 30 min at room temperature. During this process, hydrophilic copolymers were fully deposited into the entire membrane, and the PVP-VTES underwent hydrolysis reaction (2) to produce a large amount of active Si–OH. Owing to the hindrance of PET at the bottom of M0, the active hydrophilic copolymers in the membrane could only migrate unidirectionally toward the membrane’s top surface, which was in direct contact with the modifier. Subsequently, the membrane was transferred and placed in hot water at 60 °C for a period of time, during which the membrane matrix underwent interface thermal swelling. Additionally, the long-chain hydrophilic copolymer underwent thermal-crosslinking reaction (3) through active Si–OH to further form a long-chain net-like structure anchored on the membrane surface. The thickness of the modified layer could be adjusted by varying the thermal-crosslinking time. Finally, the composite membrane was naturally dried and the PET was peeled off for further experiments. The newly obtained rough surface was defined as the bottom surface (B) and the smooth side was defined as the top surface (T). The modified membrane was recorded as M1-n (n is the thermal-crosslinking time). Figure 1 shows the preparation and hydrophilic-modification process of the PVDF hydrophobic membrane and all the reaction formulas that are related to PVP-VTES; that is, the polymerization thermal-crosslinking reaction (1), the hydrolysis reaction (2), and the thermal-crosslinking reaction (3).
(1)
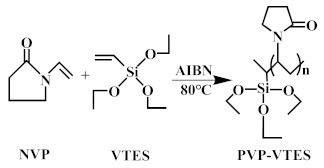

(2)
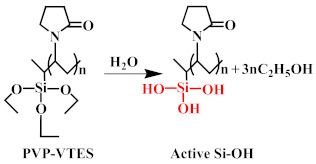

(3)
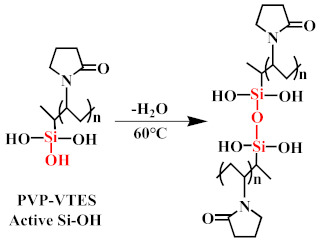


### 2.3. Membrane Characterization

Field-emission scanning electron microscopy (FE-SEM, S4800, Hitachi, Ibaraki, Japan) and confocal laser scanning microscopy (CLSM, LSM700, Zeiss, Oberkochen, Germany) were used to observe the membrane morphology and the top and bottom surface roughness, respectively. The elemental composition and chemical functional groups of both surfaces of the membrane were analyzed using X-ray photoelectron spectroscopy (XPS, Axis Ultra DLD, Kratos, Manchester, UK) with Al Kα as the radiation source and micro-Fourier transform infrared spectrometry (micro-FTIR, Cary660 + 620, Agilent, Santa Clara, CA, USA) within the range of 4000–400 cm^−1^, respectively. A universal material testing machine (Z1.0, Zwick, Ulm, Germany) was used to test the tensile strength at a rate of 50 mm·min^−1^. The pore-size distribution of the membrane was measured using a liquid-liquid pore meter (LLP-1200A, PMI, CA, USA), and Galwick was used as a wetting agent. The total porosity (*P*) of the membrane was measured via the dry and wet method; that is, the weights of the dry and wet membranes of the same sample were measured and they were used in the following formula:(4)$$P=\frac{m_{2}-m_{1}}{\rho \times A\times h},$$
where *m*_2_ (g) is the weight of the wet sample, *m*_1_ (g) is the weight of the dry sample, *ρ* (1.80 g·cm^−3^) is the density of the infiltration solution, *A* (cm^2^) is the area of the sample membrane, and *h* (cm) is the thickness. Three samples were chosen for each membrane type for testing, and the results were based on average values to minimize experimental error.

The hydrophilic layers of the membranes were dyed with Congo red aqueous solution. After drying, the membranes were fragile and kept in liquid nitrogen. They were then tested using CLSM to observe and measure the thicknesses of the hydrophilic layers (stained layers).

### 2.4. Wettability of Membrane

Wettability is an inherent feature of surfaces that is jointly controlled by surface topography and chemical composition. The surface free energy can typically be determined by the chemical composition, and the Young’s, Wenzel, and Cassie models can determine the influence of the surface topography. Moreover, the Cassie-Baxter model can explain the wetting behaviour of a porous rough structure. A fully wetted single homogeneous structure in contact with the droplets on its surfaces changes to a composite state when the roughness exceeds a particular value. The liquid cannot fill the surface completely owing to the abundant stockpile of air in the fine voids of the rough surface; thus, a practical “liquid–solid” contact [36,37] comprises “liquid-solid” and “gas-solid” contacts (Figure 2). The membrane surface droplet contacts we prepared conform to the Cassie-Baxter model equation:(5)$$cos\theta_{CB}=f_{1}\cos\theta_{1}+f_{2}\cos\theta_{2}=f_{1}\left(\cos\theta_{1}+1\right)-1,$$
where *θ*_CB_ is the surface contact angle of the composite surface, *θ*_1_ and *θ*_2_ are the intrinsic contact angles between the two media (liquid and gas) at the interface, and ƒ_1_ and ƒ_2_ are the area fractions of the liquid and gas on the solid surface (ƒ1 + ƒ2 = 1), respectively. When one of the media is air, the “gas-solid” contact angle *θ*_2_ is 180°.

The wettability of the T/B surfaces of the membrane was measured using an optical contact angle tester (OCA25, DataPhysics Instruments, Filderstadt, Germany). A 1 cm × 5 cm sample with the test side facing up was placed on a glass slide, and the water contact angle (WCA) values in air and underwater oil contact angle (OCA) values at different positions of the sample were measured. Each test used 2 μL droplets and five types of oil (soybean oil, silicone oil D5, trichloromethane, methylbenzene, and n-hexadecane) were used to test the OCA. All the WCA and OCA values were determined from the average values measured at three different points on the surface of each sample.

### 2.5. Separation of Oil-Water Emulsions

Three different wettability membranes—the hydrophobic membrane M0, Janus membrane M1, and hydrophilic membrane M1′—were subjected to the decompression separation of oil-water (O/W, W/O) emulsions. The switchable separation performance of the Janus membrane with asymmetric wetting characteristics was investigated in the separation experiments for both emulsions. First, the feed liquids (O/W or W/O emulsions) were prepared for each separation test, and the dispersed phases (water droplets and oil droplets) were dyed with Brilliant Blue and Sudan III, respectively. For the O/W emulsions, 5 g of soybean oil was added to 1 L water with SDS (0.05 g·L^−1^) as a stabilizer and vigorously stirred for 12 h. Additionally, 6 mL water and 100 mL soybean oil were stirred at high speed to prepare the W/O emulsions. The O/W-emulsion separation process was controlled by a vacuum-driven filtration system at −0.09 Mpa, and the W/O-emulsion separation pressure was −0.05 Mpa. Each type of membrane (M0, M1, M1′) was utilized for the separation of the two oil-water (O/W, W/O) emulsions three times. An ultraviolet-visible spectrophotometer (UV-2600, Shimadzu, Kyoto, Japan) was used to test the content of oil droplets in the O/W emulsions and the filtrate after the membrane separation. Moreover, the Karl Fischer moisture analyzer (831 KF Coulometer, Metrohm, Herisau, Switzerland) was used to test the concentration of water droplets in the W/O emulsions and filtrate. The water/oil flux and separation efficiency (*R*) of the membrane were calculated based on Equations (6) and (7), respectively, and the average values of each group of calculated values were used for a graph comparison:(6)$$\mathrm{Flux}=\frac{V}{S\times t}$$
(7)$$R=\left(1-\frac{C_{\mathrm{filtrate}}}{C_{\mathrm{feed}}}\right)\times 100\%,$$
where V is the volume of the filtrate (L), S is the effective area of the membrane (1.26 × 10^−5^ m^2^), T is the test time (h), and C_filtrate_ and C_feed_ are the water/oil concentrations in the filtrate and feed, respectively.

## 3. Results and Discussion

### 3.1. Morphological Structure

In this study, no additive was added to the PVDF casting solution; therefore, the apparent color of the original membrane (M0) prepared by the phase-inversion method was white. As shown in Figure 3a, the surface of M0 was flat and it had many clearly observable pores that were caused by the mutual solvent–non-solvent conversion during the phase-inversion process. Compared to the T surface, the B surface had a micro/nano-level wrinkle structure (the inset in the upper right corner is an enlarged view of the bottom surface), which was attributed to the rough template provided by the bottom non-woven fabric. The scanning image of the three-dimensional confocal microscope (Figure 3c) also showed that the surface roughness of M0 was significantly different from those of the other membranes. The roughness values were 0.591 μm and 2.928 μm, respectively, which were similar to the T/B surface morphological structural differences in the SEM image. Based on the Cassie-Baxter wetting model, a bottom surface with a higher roughness makes it easier to construct a solid-liquid-gas interface, which is an essential reason for the strong hydrophobicity of the bottom surface. As observed in the cross-section, the internal structure of the membrane presented a micro/nano “hydrangea”-like fold structure, and no structural delamination phenomenon was observed. Because no pore-causing agent was added to the casting solution, no obvious porous channels were observed in the membrane. However, the interspaces between the irregular accumulation of the “hydrangea” provided a transport channel inside the membrane.

The T/B surfaces of M1 also had significantly different structures and roughness compared to those of the other membranes, as shown in Figure 3b,d. Compared with M0 (Figure 3a,c), the morphologies of the top, bottom, and section surfaces of M1 remained unchanged. In particular, no internal structural delamination was observed in the M1 section. The porosity and mechanical properties of the membrane remained almost unchanged after modification. As shown in the test results for the average pore size, porosity, and mechanical strength of M0 and M1 in Table 1, the porosities of both membranes with the same membrane thickness were high, exceeding 70%. Compared with that of M0, the mechanical strength of M1 after the thermal-swelling treatment improved. These results indicate that the Janus membrane prepared via the hydrophilic modification steps, namely the hydrolysis, offset, and thermal-crosslinking polymerization of the hydrophilic polymer in the membrane, was steady, and no delamination in the internal membrane structure, blockage of membrane pores, or porosity reduction was observed.

The thermal-crosslinking time is the key factor in the hydrophilic-modification process of the PVDF hydrophobic membrane. If the time exceeds the boundary, the hydrophilic copolymer will completely occupy the space inside the membrane and anchor inside the entire membrane after thermal crosslinking; finally, a purely hydrophilic membrane is obtained—that is, the top and bottom surfaces of the membrane are hydrophilic. Congo red aqueous solution was simultaneously added to the T surface of M1 and M1′ for dyeing. The solution quickly spread and completely infiltrated the hydrophilic layer of the modified membrane. Therefore, the thickness of the dye layer (i.e., the hydrophilic layer) and the dividing line between the hydrophilic (red) and hydrophobic layers (gray-white) could be observed using a laser microscope, as shown in Figure 3e. Compared with M0 with a similar thickness, M1′ exhibited red dye throughout the whole section, which was consistent with its purely hydrophilic property. A color-dividing line was clearly observed on the M1 section surface, which proved the existence of a hydrophilic layer with a particular thickness and intuitively showed the special integrated structure of the Janus membrane, which has a hydrophilic layer/hydrophobic layer without delamination. The hydrophilic layer thickness of the Janus membrane is related to the membrane structure, hydrolysis time, and thermal-crosslinking time during the hydrophilic modification. M1 in this study refers to the modified membrane with a thermal-crosslinking time of 6 h.

### 3.2. Chemical Composition

The data obtained from the ATR-FTIR and XPS characterizations showed that the chemical elements of the final modified membrane were significantly different from those of the original hydrophobic membrane. As shown in Figure 4a, the characteristic peak of the C=O bond in the hydrophilic polymer PVP-VTES appeared at 1655 cm ^−1^ in M1, and the peaks at 840 cm^−1^ and 3500 cm^−1^ were attributed to the Si-O-Si bond in PVP-VTES and the Si-OH bond formed by the partial hydrolysis of PVP-VTES, respectively. As shown in Figure 4b and Table 2, the N, O, and Si elements in PVP-VTES appeared in M1; however, M0 exhibited only the C and F elements. The ATR-FTIR and XPS diagrams show that the bottom surface of M1 also contained three elements: N, O, and Si. This is because after the original membrane was placed in the hydrophilic modifier, the hydrophilic polymer entered the membrane from the membrane surface and reached the bottom of the membrane through the intermembrane gap. However, a comparison of the ATR-FTIR and XPS diagrams of the Janus membrane top/bottom surfaces shows that the peak intensity of the infrared characteristic of the C=O and Si–O–Si bonds on the bottom was relatively weak, and the signal intensity of Si or N in PVP-VTES was also clearly weak at the bottom of M1. As shown in Table 2, the contents of O (1.5%), N (8.2%), and Si (1.4%) on the B surface of M1 were significantly lower than those on the T surface (3.6%, 13.1%, and 2.3%, respectively). The difference in the elemental content between the two sides of M1 is correlated with the unidirectional migration of the hydrophilic copolymer. The hydrophilic polymer could only migrate upward and accumulate on the top surface owing to the barrier of the support body at the bottom of M1, which resulted in a concentration gradient of hydrophilic copolymers inside the membrane and a considerably stronger hydrophilicity on the top surface. However, the hydrophobicity of the M1 bottom surface was formed under the combined effect of the low surface energy of the PVDF substrates and the micro/nano rough structure caused by stripping the non-woven fabric.

If the underside support of the original membrane was first stripped, and the same hydrophilic modification was then carried out, a purely hydrophilic membrane (M1′) could be obtained, even with a short thermal-crosslinking time. This is because the hydrophilic modifier can easily be immersed in the membrane from the underside. PET has three important functions in the preparation and modification of the original membranes. First, it acts as a supporting body during the membrane preparation. Second, a rough template is provided from PET, and its stripping can create a micro/nanostructure of the bottom surface, thereby ensuring the hydrophobic performance of the Janus membrane bottom surface under the collaborative effectiveness of low surface energy and the micro/nano rough structure. Finally, PET plays a barrier role in the hydrophilic modification experiment; thus, the hydrophilic modifier can only be immersed and migrated on the top side and enriched on the membrane surface, which ensures the hydrophilic property of the Janus membrane surface.

### 3.3. Asymmetric Wettability

To characterize the opposite wettability of the Janus membrane, the WCA in air and the underwater oil contact angle were measured and compared with those of the hydrophobic membrane. Figure 5a,b shows the variation trend of the WCA values on the T and B surfaces of the membrane over time. The initial WCA values of the T and B surfaces of M0 were 112° (b) and 145° (a), respectively, and both of these values did not decrease considerably within 2 min, which proved that both sides of M0 exhibited good hydrophobicity. The higher WCA value of the B surface also proved that the micro/nanostructure was a crucial reason for the improved hydrophobicity. Compared with that of M0, the WCA of the M1 underside was 142°, which also did not change significantly within 2 min. Figure 3b,d shows that under the synergistic effect of low surface energy and a rough structure, the M1 underside had a hydrophobicity similar to that of M0. However, the WCA value of the M1 surface decreased significantly in a short period of time, from 70° to 0°, and the water droplets completely spread out and wet the membrane surface, thereby proving that M1 surface had a good hydrophilic property. As shown in Figure 5c–g, five types of oil (including three kinds of light oil: soybean oil, methylbenzene, and n-hexadecane, and two types of heavy oil: silicone oil D5 and trichloromethane) were used to determine the underwater oil contact angles on the T surfaces of M0 and M1. The underwater OCA values of methylbenzene, n-hexadecane, and trichloromethane on the T surface of M0 rapidly decreased to 0° in approximately 10 s. The soya oil and the silicone oil D5 slowly spread out on the T surface owing to their viscosities; however, they eventually completely spread out, and the OCA values decreased to 0°. This was due to the good hydrophobicity of M0, which easily produced hydrophobic interaction forces with oil droplets, thereby enabling them to be quickly adsorbed on the hydrophobic surface. The underwater OCA values of various oil droplets on the T side of M1 were higher than 140°, compared with those of M0 (Figure 5c–e). Moreover, the OCA values of silicone oil D5 and trichloromethane were superhydrophobic (>150°, Figure 5f,g), and the contact angle remained stable. This is because the hydrophilic layer of M1 was completely infiltrated underwater; thus, the adhesion of oil droplets on the T surface was low, and a hydrophilic surface (super)oleophobic effect was achieved. In addition, compared with other references, as listed in Table 3, the differences of the contact angle between the surface and the underside of the Janus membrane are generally higher. In summary, the results showed that the internal hydrophilic modification of the PVDF hydrophobic membrane was successful, and the obtained modified membrane had good top-surface underwater (super)oleophobicity and bottom hydrophobicity. A Janus membrane with this opposite wetting property can be used for the switchable separation of O/W and W/O emulsions.

### 3.4. Hydrophilic Layer Thickness Control

In the hydrophilic modification experiment of the hydrophobic membrane, the PVP-VTES long chains were deposited in the membrane, the thermal-crosslinking reaction between the long chains was carried out via active Si–OH, and the network structure was further formed and anchored on the membrane surface. Due to the existence of the non-woven fabric at the bottom of the original membrane, the hydrophilic copolymer with easy hydrolytic activation can only migrate to the top surface of the membrane unidirectionally; thus, the concentration of the hydrophilic copolymer deposited in the membrane showed a gradient trend. Therefore, after the thermal-crosslinking reaction, the thickness of the hydrophilic layer increased with increasing thermal-crosslinking time. As shown in Figure 6, a distinct red layer (i.e., the hydrophilic layer) was displayed on the M1 section, compared with M0, after staining the hydrophilic surface using Congo red, and its thickness increased with the increasing thermal-crosslinking time.

The hydrophilic layer thicknesses of the modified membranes under different thermal-crosslinking times were tested separately, the percentage value of the hydrophilic layer thickness of each sample compared to its total membrane thickness was calculated, and the average number of values was obtained, as shown in Table 4. After deposition in the hydrophilic modifier for 30 min and thermal crosslinking in hot water for 1 h, the thickness of the hydrophilic layer was 30.49 μm, and it comprised 15.25% of the total membrane thickness. When the thermal-crosslinking time was extended to 18 h, the hydrophilic layer thickness increased significantly (to 99.55 μm), and it comprised half of the total membrane thickness (50.07%). When the thermal-crosslinking time continued to be prolonged, a purely hydrophilic membrane (M1′) was obtained, and its thickness was completely composed of the hydrophilic layer.

The hydrophilic layer thickness is an important factor that affects the separation of oil-water (O/W, W/O) emulsions. Therefore, M0, M1′, and M1-6 h were selected for emulsion-separation and purification experiments and the oil/water flux and separation efficiency of the three membranes were compared.

### 3.5. Oil–Water Emulsions Separation

Oil–water (O/W, W/O) emulsion decompression separation experiments were performed on M0, M1′, and M1-6 h (Figure 7a). The switchable separation properties of the Janus membranes with asymmetric wetting characteristics are discussed. For the O/W emulsions, membrane-separation experiments were conducted on the T/hydrophilic sides of the membranes facing the feed liquid, and for the W/O emulsion, the B/hydrophobic sides of the membranes were used to face the feed liquid. As shown in Figure 7b, the filtrate after membrane separation and emulsions before separation exhibited clear differences in color, and the filtrate of M1 was the lightest and had the lowest turbidity. The separation effects of the three membranes on the oil-water (O/W, W/O) emulsions were different, and the separation effect of M1 was relatively better than those of the other membranes.

Figure 7c,d compares the water/oil flux and separation efficiency of the three membranes with different wettabilities. For O/W emulsions, the top surface of M0 exhibited hydrophobicity; therefore, water could not infiltrate the surface of the membrane, and its water flux was only 2.52 L/m^2^·h (Figure 5b). The purely hydrophilic membrane M1′ had a higher water flux because of the hydrophilicity of the membrane surface; however, its separation efficiency was only 30.68% because the water wrapped around some oil droplets and passed through the membrane together.

When the hydrophobic bottom surface of M0 treated the W/O emulsions, the oil wrapped around some small molecules of water and was rapidly adsorbed on the hydrophobic surface and across the membrane pores. A higher oil flux was eventually gained; however, the separation efficiency was relatively low. The purely hydrophilic membrane exhibited the opposite results to those of M0. The entire membrane had a strong hydrophilic nature, which prevented the passage of oil, and eventually, almost no oil flux was observed for W/O emulsions (0.89 L/m^2^·h). The experimental results showed that purely hydrophobic or hydrophilic membranes are not beneficial for the separation of oil-water (O/W, W/O) emulsions because of their low flux or low separation efficiency.

In contrast to M0 or M1′, M1 had a high flux and separation efficiency, as shown in Figure 7c,d. When separating O/W emulsions, water easily infiltrated and penetrated the membrane because the T surface of the Janus membrane was enriched with hydrophilic copolymers (Table 2). Moreover, the T surface also exhibited high underwater lipophobicity, and the contact angles of various oils underwater were approximately 150° (Figure 5). Therefore, the T surface prevented the adsorption of oil droplets. Second, when water and a small amount of oil entered the hydrophilic layer under the action of pressure to reach the hydrophilic-hydrophobic interface, the presence of the hydrophobic layer increased the residence time of the oil-water mixture in the hydrophilic layer, thereby promoting the demulsification of the stabilized emulsions and the aggregation and merger of small oil droplets. Consequently, more oil droplets were trapped on the surface, whereas the water passed smoothly through the hydrophobic layer under pressure. Therefore, the separation flux of the O/W emulsions was 22.88 L/m^2^·h and the separation efficiency was up to 93.35%, as shown in Figure 7c.

Similarly, the bottom surface of the Janus membrane exhibited both good hydrophobicity and lipophilicity. Thus, it can be used for the separation of W/O emulsions. The continuous phase in the W/O emulsions was rapidly adsorbed and it formed an oil layer on the bottom surface of the membrane to achieve a good water-repellent effect. Moreover, the presence of the hydrophilic layer promoted the demulsification of the stabilized emulsions. More water droplets aggregated and were trapped on the surface, whereas the oil passed smoothly through the membrane under pressure. As a result, the separation flux of the W/O emulsions was 17.45 L/m^2^·h and the separation efficiency was as high as 91.47%.

### 3.6. The Membrane-Stability Test

The reproducibility and stability of the modified membranes were verified via cycling experiments. Figure 8 shows the changes in the flux and separation efficiency of the M1 membrane after five consecutive cycles of filtration in an oil-in-water emulsion (soybean, petroleum ether, and n-hexane). At a pressure of 0.09 MPa, the M1 membrane was able to maintain a high flux and retention rate after several cycles of the experiment in different oil-in-water emulsions. This demonstrates that the prepared Janus membranes have excellent antifouling properties and long-term stability, which are crucial for practical applications.

### 3.7. Separation Mechanisms

The hydrophilic layer thickness is one of the main influencing factors when separating oil-water (O/W, W/O) emulsions with a Janus membrane. Based on the change in the thickness of the hydrophilic layer in Figure 9, when the thickness of the hydrophilic layer was extremely small (purely hydrophobic membrane), the oil flux was large when separating the W/O emulsions; however, a high water-rejection rate was not guaranteed [42]. The emulsion-separation results were completely opposite to those of the purely hydrophobic membrane when the hydrophilic layer thickness was extremely large (purely hydrophilic membrane); the membrane exhibited a large water flux and low oil-repulsion rate when separating O/W emulsions. However, a Janus membrane with a particular hydrophilic layer thickness may achieve comparatively good flux and separation efficiency. Therefore, a good separation effect can be obtained by adjusting the thickness of the hydrophilic layer via the thermal-crosslinking time.

When using hydrophilic membranes to separate oil-in-water emulsions, large oil droplets were repelled by the membrane pores outside the membrane, due to the uneven droplet size of the emulsion, whereas small oil droplets entered the membrane with the flow of water. Therefore, the retention rate of the oil droplets was relatively low. Similarly, when treating water-in-oil emulsions with hydrophobic membranes, small droplets passed through the membrane into the filtrate. When the hydrophilic layer of the Janus membrane was used upward to treat oil-in-water emulsions, the sieving effect of the membrane pores caused large oil droplets to be repelled outside the membrane. However, water flow and small oil droplets entered the hydrophilic layer and were blocked by the hydrophobic layer below, where the small oil droplets aggregated into large oil droplets. After breaking the emulsion, large oil droplets were discharged from the membrane by the hydrophilic layer, and the water flow passed through the hydrophobic layer under negative pressure. Therefore, the Janus membrane had a high oil-droplet retention rate. Similarly, when treating water-in-oil emulsions with Janus membranes, the retention rate of the Janus membranes was higher than that of hydrophobic membranes, due to the combined effects of membrane-pore sieving and emulsion breaking.

## 4. Conclusions

This study proposed a preparation strategy for Janus membranes, a hydrophilic layer established inside a hydrophobic matrix membrane. The proposed preparation method for the Janus membrane is simple, the membrane has no structural stratification or decrease in porosity, and the hydrophilic layer thickness is controllable and adjustable. The final modified membrane with completely opposite wettability may be used for the switchable separation of oil–water (O/W, W/O) emulsions. The separation flux of O/W emulsions on the hydrophilic surface of the Janus membrane was 22.88 L/m^2^·h and the separation efficiency was up to 93.35%. Moreover, the separation flux of W/O emulsions on the bottom surface was 17.45 L/m^2^·h and the separation efficiency was as high as 91.47%. Compared with the low flux and separation efficiency of the purely hydrophobic and hydrophilic membranes, the Janus membrane exhibited high flux and separation efficiency for both oil–water emulsions, which will be a future research direction with potential application value.

## Figures and Tables

**Figure 1 membranes-13-00415-f001:**
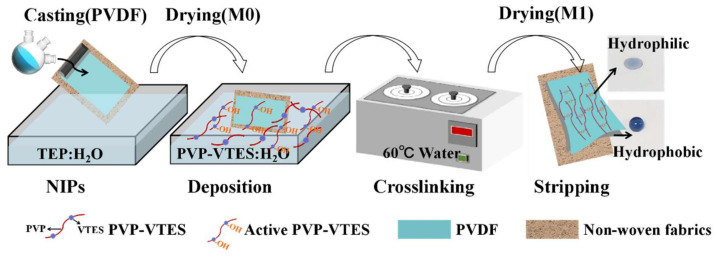
Schematic of the preparation of M0 and M1.

**Figure 2 membranes-13-00415-f002:**
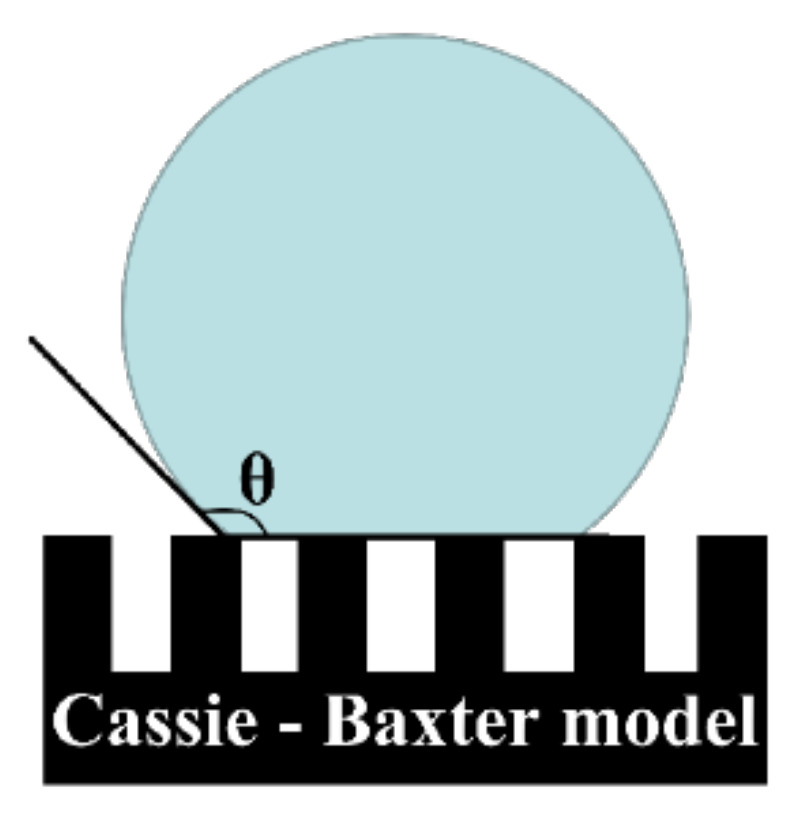
The Cassie-Baxter wetting model.

**Figure 3 membranes-13-00415-f003:**
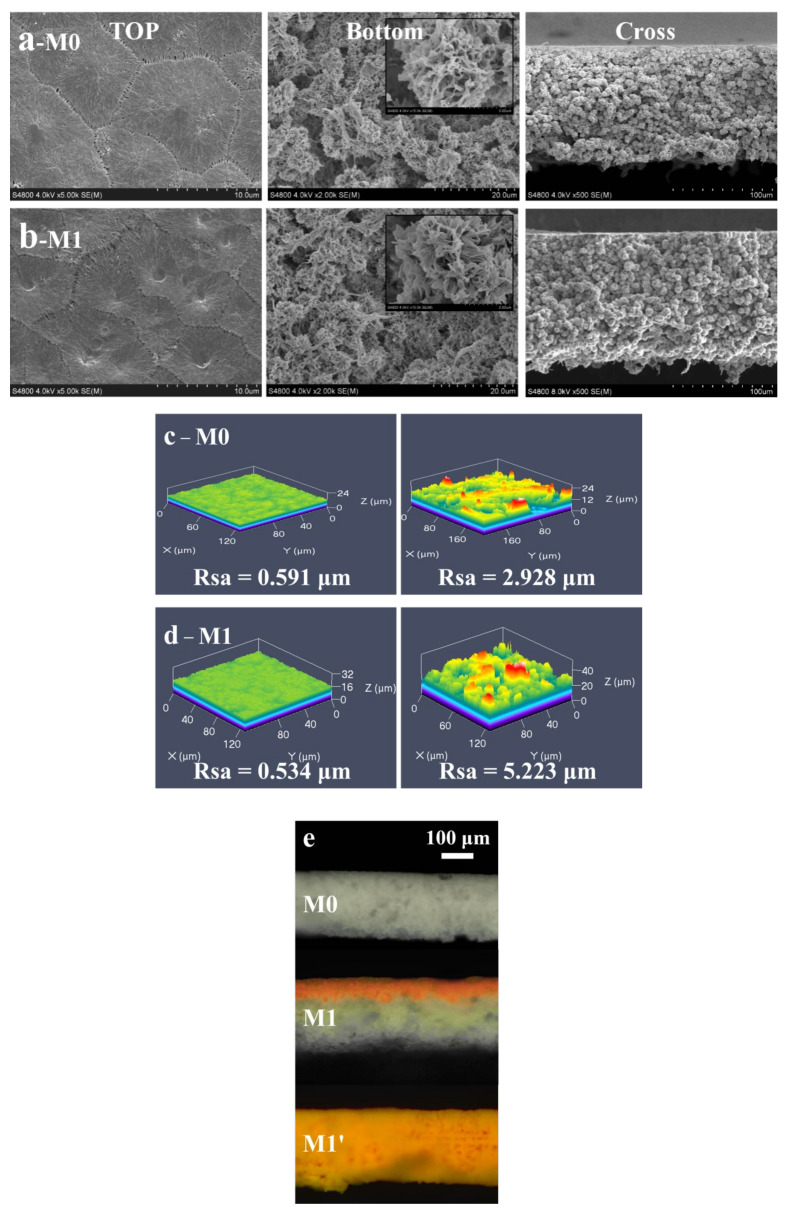
The morphological structure of membranes: (**a**,**b**) are the morphological structures of the top surface, bottom surface, and section surface of M0 and M1, respectively; (**c**,**d**) show the roughness of the top and bottom surfaces of M0 and M1, respectively; (**e**) displays the stained cross-section of M0, M1, M1′, and the red layer with a clear thickness is the hydrophilic layer.

**Figure 4 membranes-13-00415-f004:**
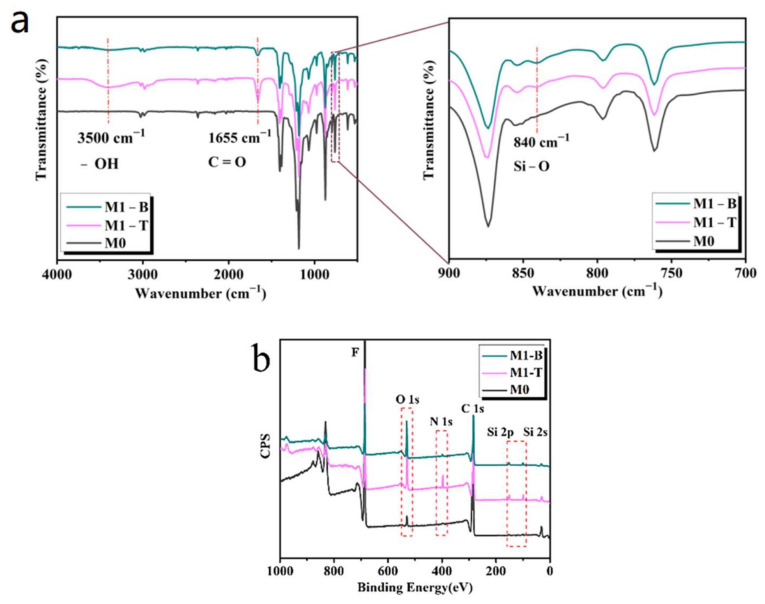
Chemical analysis of M1 and M0: comparisons of (**a**) chemical groups and (**b**) constituent elements for the top (T) and bottom (B) surfaces of M1 and M0.

**Figure 5 membranes-13-00415-f005:**
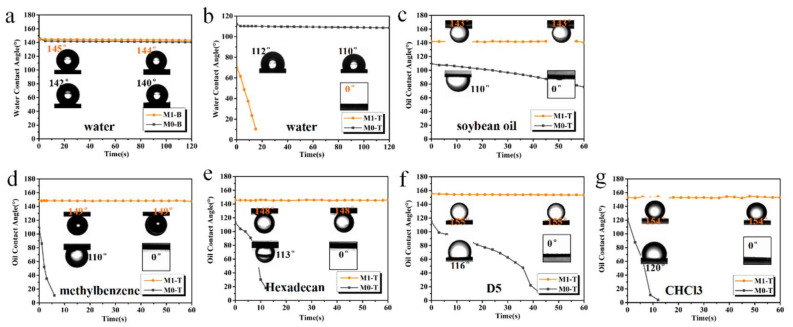
The surface wettability of membranes: (**a**,**b**) are the changes and comparisons of the WCA values in the air on the bottom (B) and top (T) surfaces of M0 and M1, respectively; (**c**–**g**) are the changes and comparisons of underwater OCA values of soybean oil, methylbenzene, n-hexadecane, silicone oil D5, and trichloromethane on the T surfaces of M0 and M1, respectively.

**Figure 6 membranes-13-00415-f006:**
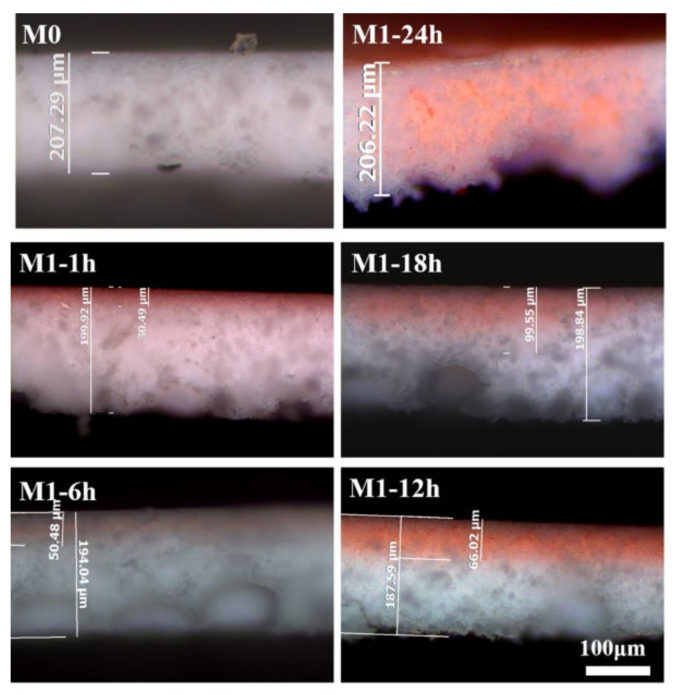
The controllable hydrophilic layer thickness: compared with M0, the M1 section shows a significant red layer, and its thickness increases with an increase in the thermal-crosslinking time.

**Figure 7 membranes-13-00415-f007:**
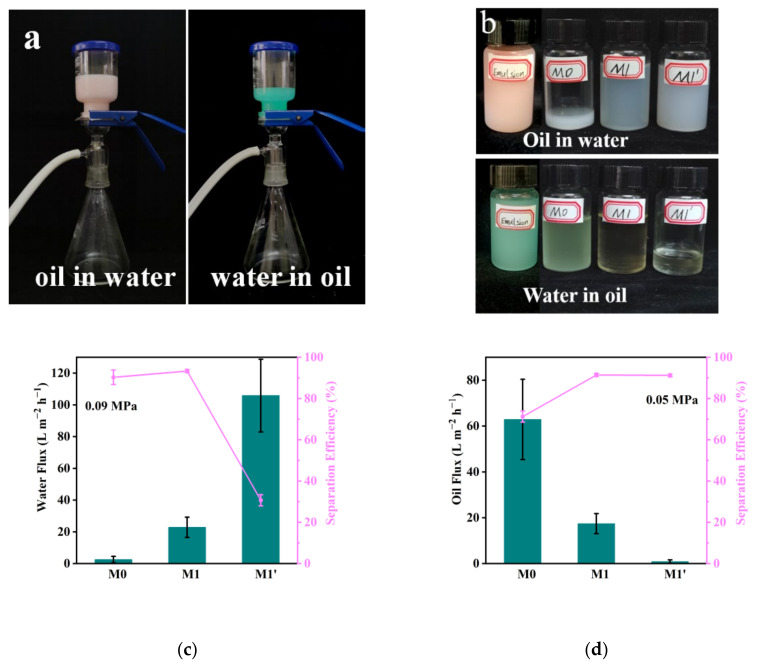
The switchable oil–water (O/W, W/O) emulsion separation test of M0, M1, and M1′: (**a**) the device of emulsion decompression separation; (**b**) the emulsions and filtrates after decompression separation via three types of membranes; (**c**,**d**) water flux and oil flux as well as separation efficiency of the three types of membranes, respectively.

**Figure 8 membranes-13-00415-f008:**
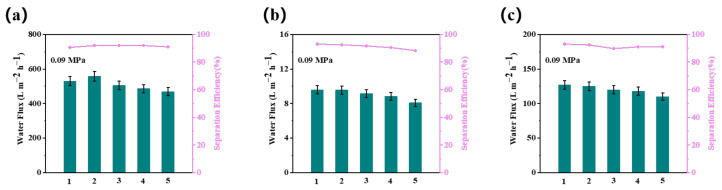
The cycle flux and separation efficiency of M1 when separating an oil-in-water emulsion of (**a**) soybean, (**b**) petroleum ether, and (**c**) n-hexane.

**Figure 9 membranes-13-00415-f009:**
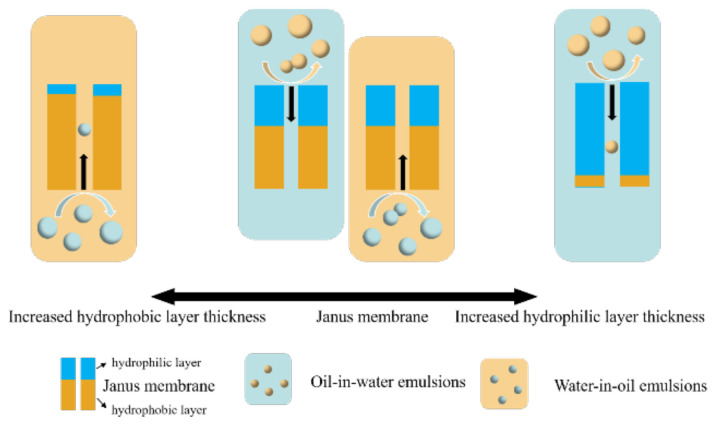
The influence of hydrophilic layer thickness and the mechanism of the Janus membrane.

**Table 1 membranes-13-00415-t001:** Pore size, porosity, and tensile strength of M0 and M1.

Membrane	Thickness (μm)	Mean Pore Size (μm)	* Porosity (%)	Tensile Strength (MPa)
**M0**	200 ± 10	0.38	74.71	0.85
**M1**	200 ± 10	0.33	73.26	1.16

* Porosity (%): measured by dry–wet method.

**Table 2 membranes-13-00415-t002:** Chemical element content on membrane surfaces of M0 and M1.

At (%)	C	F	N	O	Si
M0	50.1	49.9			
M1-T	60.6	20.4	3.6	13.1	2.3
M1-B	56.4	32.5	1.5	8.2	1.4

**Table 3 membranes-13-00415-t003:** Comparison of the contact angle of the Janus membrane with other modified materials reported in literature.

Literatures	Top Surface Angle (θ_1_, deg)	Bottom Surface Angle (θ_2_, deg)	Δθ (θ1–θ2, deg)
Functional cotton fabric with single-faced superhydrophobicity [38]	140	40	100
Surface segregation of fluorinated modifying macromolecule for hydrophobic/hydrophilic membrane [39]	94	78	15
PET/PTFE@TA-DETA multifunctional Janus membranes [40]	115	16	99
PDA/SWCNT bilayer membranes [41]	104	48	56
This study	0	142	142

**Table 4 membranes-13-00415-t004:** Variation trend of hydrophilic layer thickness of Janus membrane.

Modification Time (h)	Total Membrane Thickness (μm)	Hydrophilic Layer Thickness (μm)	Hydrophilic Layer Percentage (%)	The Mean Percentage (%)
1	199.92	30.49	15.25	15.57 ± 2.71
6	194.04	50.48	26.02	23.66 ± 2.75
12	187.59	66.02	35.19	37.94 ± 2.55
18	198.84	99.55	50.07	54.18 ± 6.09
24	206.22	206.22	100.00	100.00

## Data Availability

Not applicable.

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
