# Peer review of "Preparation and Characterization of a Janus Membrane with an “Integrated” Structure and Adjustable Hydrophilic Layer Thickness"

_membranes, 2023, doi:10.3390/membranes13040415_

Round 1

Reviewer 1 Report

The manuscript titled "Preparation and characterization of Janus membranes with "integrated" structure and adjustable hydrophilic layer thickness" describes the synthesis of a Polyvinylidene fluoride hydrophobic matrix membrane modified by hydrophilic polymer, poly(vinylpyrrolidone-vinyltriethoxysilane) (PVP-VTES) with asymmetric wettability. The manuscript discusses the morphological structure, chemical composition, wettability, hydrophilic layer thickness, and porosity of the membranes using standard characterization techniques. The membrane was successfully tested for switchable separation of oil-water emulsions. The manuscript is well arranged and appropriately cited. However, there are a few reviewer comments as follows:

1.      The reviewer suggests a revision of the manuscript in terms of grammatical and typo errors.

2.      The membrane showed lower separation efficiencies than in other reported literature. Appropriate justification is required.

3.      The reviewer suggests the addition of a comparative table.

Author Response

Point1. The reviewer suggests a revision of the manuscript in terms of grammatical and typo errors.

Response 1: 

We apologize for the poor language of our manuscript. We have now worked on both language and readability and have also involved native English speaker for language corrections. We really hope that the language level have been substantially improved.

Point2. The membrane showed lower separation efficiencies than in other reported literature. Appropriate justification is required.

Response 2: 

We thank the reviewer for pointing out this issue. We indeed showed lower separation efficiencies than other reported literature. This may be due to the low sensitivity of the instrument used to bias the measurement results. However, compared with the original membrane, the retention rate of the modified membrane has been greatly improved. Later we will use the total organic carbon instrument to reduce the bias.

Point3. The reviewer suggests the addition of a comparative table.

Response 3:

We will be happy to edit the text further based on helpful comments from the reviewers. We have already added the comparision table in the manuscript.

Reviewer 2 Report

The article describes the preparation and characterization of Janus membranes with an "integrated" structure and adjustable hydrophilic layer thickness.

The theme is interesting, although some concerns must be addressed:

-          Please remove the underline from the word “increasing” (the “Introduction” section”).

-          Please try to emphasize the aim and novelty of the study at the end of the ‘Introduction” section.

-          Full information about the reagents and used instruments (company, city, country) must be provided.

-          All figures must be included after their first mention in the text (not before as fig 1 was introduced).

-          Please verify the equations so that they are written with the Equation Editor.

-          Please increase the dimensions and resolution of Figure 4 and Figure 6.

-          The English language must be slightly revised.

Author Response

Point 1: Please remove the underline from the word “increasing” (the “Introduction” section”).

Response 1: Thank you for point out this problem. We have removed underline of the word.

Point 2: Please try to emphasize the aim and novelty of the study at the end of the ‘Introduction” section.

Response 2: Thank you for the above suggestion. According to the revised content, we added the illustration of aim and novelty of the study.

Point 3: Full information about the reagents and used instruments (company, city, country) must be provided.

Response 3: Thank you for your comment. We completed the full information about the reagents and instruments.

Point 4: All figures must be included after their first mention in the text (not before as fig 1 was introduced).

Response 4: Thank you for your rigorous comment. We have checked and fixed this error.

Point 5: Please verify the equations so that they are written with the Equation Editor.

Response 5: Thank you for point out this problem. We have rewritten the equations by the Equation Editor.

Point 6:  Please increase the dimensions and resolution of Figure 4 and Figure 6.

Response 6: Thank you for your comment. We have increased the dimensions and resolution of pictures.

Point 7: The English language must be slightly revised.

Response7: We are very sorry for the mistakes in this manuscript and inconvenience they caused in your reading. The manuscript has been thoroughly revised and edited by a native speaker, so we hope it can meet the journal's standard. Thanks so much for your useful comments.